# Information technology capability, open technological innovation and firm growth

**Weizhi Yao**[ID]**[1,2,3]\*, Lianshui Li[1,3,4]**

**1** School of Economics and Management, Southeast University, Nanjing, China, **2** School of Business and Management, Queen Mary University of London, London, England, United Kingdom, **3** School of Business, Wuxi Taihu University, Wuxi, China, **4** School of Management Engineering, Nanjing University of Information Science and Technology, Nanjing, China

\* yaowz@seu.edu.cn

**Data Availability Statement:** All relevant data are within the paper and its Supporting Information files.

**Funding:** This work was supported by the National Natural Science Foundation of China Youth

## Abstract

The aim of this paper is to investigate the effects of information technology (IT) capability on firm growth in the context of open technological innovation. The paper utilized a logical deductive approach to develop hypotheses and analytical frameworks, and collected empirical data from 256 Chinese new ventures. Regression analysis and structural equation models were used to test the hypotheses and analyze the data. The results showed that IT capability, including flexibility and integration of information technology, significantly influenced firm growth, and open technological innovation partially mediated the relationship between IT flexibility and firm growth, and significantly mediated the relationship between IT integration and firm growth. The paper's limitations include the cross-sectional design, limited sample size, and potential unobserved variables such as organizational learning that could affect the relationship between IT capability and firm growth. The research is the first to investigate the effects of IT capability on firm growth based on the mediation of open technological innovation in China, contributing to the literature on IT capability and providing insights for managerial practice in the sharing economy era.

## 1. Introduction

In 2021, China adopted and implemented the "14th Five-Year Plan" to guide its economic development and the growth of firms over the next five years. This plan identifies "information technology," "innovation," and "entrepreneurship" as key drivers of economic growth, and places a strong emphasis on encouraging business innovation and supporting the formation and growth of new firms. Encouraging self-employment has emerged as a primary strategy to address the talent employment problem and promote stable economic growth and market economic vitality. In this context, new ventures play a crucial role in providing employment opportunities and promoting technological innovation, as well as contributing to social and economic development [1].

Despite their potential, new ventures often face challenges in acquiring external resources due to their "newness," lack of full performance records, and information asymmetry, which can exacerbate resource constraints and create uncertainty in their growth trajectory [2, 3].

Program, Research on the Intelligent Transformation and Upgrading of China's Manufacturing Industry based on Data Integration (Grant 72102061) awarded to LL. The funder had no role in study design, data collection and analysis, decision to publish, or preparation of the manuscript.

**Competing interests:** The authors have declared that no competing interests exist.

Academic research has explored various theories and strategies to address these challenges, including resource integration and patching [4, 5]. However, new ventures still struggle to meet their growth demands with internal resources alone, often requiring deep relationships with external organizations and resource repurposing to drive innovation [6]. Additionally, new ventures must prioritize innovation to break existing corporate monopolies and adapt to rapidly changing external circumstances, thereby minimizing the risks associated with their relative lack of experience and expertise [7].

In the current era characterized by the Internet and the digital economy, technology innovation is accelerating, and demand is rapidly evolving. The complexity, systematicity, timeliness, and high investment required for innovation have made a significant impact on innovation agents. However, with the advancement of Internet technology and the transformation of corporate innovation concepts, new ventures can aggregate innovation resources worldwide by building information technology capabilities. This approach breaks the limits of traditional ownership and achieves low cost and high efficiency in innovation at a fast pace. Open technological innovation involves organizations that go beyond their original organizational boundaries to tap into external sources of innovation knowledge. They can effectively integrate these resources through internal organizational processes, which enables them to turn them into technological innovation achievements that add value [8].

Organizational innovation openness is considered a fundamental component for determining the success of enterprises in open technological innovation research. Meanwhile, the industrial environment is changing rapidly, and new ventures face new challenges as a result of the rapid growth of internet-based information technology [9]. The economic environment has dramatically changed, and innovation efforts have become more complex, with a clear cross-domain tendency. Firms are struggling to meet the new requirements with their existing resources, and using external resources for open innovation has become a significant consideration for them [10]. Several factors, such as perceived cost savings and income generation, external pressure, organizational preparedness, and perceived ease of use, have a significant impact on IT investments for new ventures [11, 12]. In new ventures, IT investments may vary from IT investment in big enterprises since a smaller number of people have decision-making responsibilities, standard procedures are not created, and long-term planning is restricted. Furthermore, there is a higher dependency on external IT professionals in new ventures [11]. However, IT capability may help new ventures survive in the long run by providing access to external knowledge and financial resources, building trust and legitimacy through widespread information transmission, and improving social network links [13]. New environmental conditions also enable new ventures to overcome resource constraints in the innovation process by improving their IT capability to achieve a long-term competitive advantage [14].

Previous research has largely focused on the impact of resource constraints on the growth of new ventures, which is a crucial concern [15, 16]. However, while resource constraints are undoubtedly a significant challenge for new ventures, there may be other factors that affect the performance and growth of firms that have not been fully explored. One of these factors could be the relationship between IT capability and performance. There have been several studies investigating the link between IT capability and firm performance, but the results have been inconclusive. Some studies have shown a positive relationship between IT capability and firm performance [14, 17], while others have reported a no relationship [18]. These mixed results suggest that there may be some missing links in the relationship between IT capability and firm performance that have not been fully explored.

Therefore, this study aims to examine the influence of open technological innovation, which is one of the potential mediators, in the relationship between IT capability and firm growth. Open technological innovation refers to the use of external knowledge and resources

to develop new products, services, and processes. The study will investigate whether open technological innovation can serve as a mediator between IT capability and firm growth, filling the gap in the existing literature.

This study contributes to the literature in three ways. Firstly, it explores the relationship between IT capability and firm growth, expanding the current research on firm growth that has primarily focused on IT investments. Secondly, the study proposes a conceptual model that considers the mediating variable of open technological innovation in this relationship. Lastly, this research is significant because while previous innovation management research has primarily focused on advanced economies, it provides insight into China, which has yet to be extensively explored.

The structure of the paper is organized as follows. Section two provides a literature review and develops the hypotheses. Section three shows the data and methods employed in the empirical study. Section four presents the empirical results, and Section five concludes with some important contributions and limitations of the study and directions for future research.

## 2. Literature review and hypotheses development

### 2.1 IT capability and firm growth

With the continuous development of internet technology, a rising number of firms will encounter environmental challenges and will need to employ IT to gain a competitive advantage. Firms can use their internal and external resources more efficiently through IT solutions to promote their growth. However, some research has indicated the existence of the so-called "IT productivity paradox," in which a company's high investment in information technology has not brought the expected benefits for the enterprise [19]. In order to further understand the causes of the "IT productivity paradox," this study also provides a framework based on capabilities with which to understand IT capability with the integration of different sources of knowledge. The RBV (Resource-based View), as proposed by Barney (1996), posits that a firm's unique resources and capabilities are the primary sources of its competitive advantage [20]. These resources and capabilities may include a firm's physical assets, human capital, technology, and organizational processes, among others. The RBV framework assumes that these resources are heterogeneous and may be difficult to replicate by competitors, leading to a sustained competitive advantage. The Dynamic Capabilities framework, as developed by Peteraf et al. (2013) and Teece and Leih (2016), builds on the RBV by emphasizing a firm's ability to adapt to changing environments through continuous learning and innovation [21, 22]. The Dynamic Capabilities perspective suggests that firms can develop new resources and capabilities by leveraging their existing ones, and by continuously adjusting their internal and external activities to respond to changing market conditions.

According to information system researchers, competitors may readily imitate IT resources by acquiring similar hardware and software, so IT resources alone cannot provide firms with long-term competitive advantages [23]. The unique IT capability of enterprises is what truly creates IT competitive advantages [24]. Regarding that, empirical research on the relationship between IT capability and firm performance revealed that firms can achieve excellent firm performance from information technology if they can effectively integrate and use IT resources to create unique IT capabilities, confirming the importance of developing effective IT capabilities to achieve outstanding firm performance. Firms are increasingly reliant on information technology to improve their information gathering and processing skills, better implement marketing strategies or innovate new ones, and maintain competitive advantages in an increasingly complicated and changing market environment.

Based on our understanding of information technology capability constitution in this study, we believe that heterogeneous IT capabilities are crucial elements in developing a fundamental competitive advantage for firms, especially for new ventures. An organization's IT capability reflects the characteristics of integrated IT environments and is defined as an organization's ability to integrate all of its available IT resources [17]. Furthermore, IT capabilities help new ventures to effectively manage their information sources, better react to environmental changes, and grab possible opportunities by improving information governance and control, which reflects the flexibility of IT capability [25]. Therefore, IT capability primarily refers to the integrated and flexible use of IT and other resources, which allows businesses to effectively utilize internal and external information to gain a competitive advantage and improve performance. The key for firms to improve performance through IT is to leverage the complementarity of IT capability and resources [26]. Firms that can creatively combine IT resources and develop heterogeneous IT capability are more likely to achieve long-term development [14]. However, some studies have found that IT capability has no significant relationship with firm performance [18]. In addition, some studies found that while IT capability may not directly improve business performance, it does indirectly promote firm competitiveness by establishing a complex organizational value chain that, in response, provides a competitive advantage [27]. Further study on IT capability from the micro-mechanism level is needed to properly explain this phenomenon.

Based on previous studies, this research explores two dimensions of the IT capability of new ventures, specifically, IT flexibility and IT integration [28]. IT flexibility refers to a company's capacity to adjust rapidly to changing demands and facilitate knowledge exchange between internal and external sources using information technology. Firms may use this capability to integrate various resources, meet knowledge innovation needs, and increase knowledge reorganization efficiency to promote innovation performance [29]. IT integration refers to a company's capability to use an Internet information system to access, share, summarize, and integrate information from both internal and external sources, providing for much more rapid and effective knowledge sharing between firms and innovation partners. This capability illustrates the intimate relationship between internal and external information systems, indicating that connection helps firms growth in collaboration with their innovative partners [30]. Therefore, we propose the following hypotheses:

*H1*. IT capability has a positive effect on the growth of new ventures.

*H1a*. IT flexibility has a positive effect on the growth of new ventures.

*H1b*. IT integration has a positive effect on the growth of new ventures.

## 2.2 IT capability and open technological innovation

IT is becoming increasingly significant in promoting economic and social growth, since it supports open technology innovation while also providing necessary technical assistance [28]. New ventures are focusing their studies on how to efficiently use IT to promote open technical innovation [31]. Technological progress in the framework of open technological innovation is primarily done through internal and external research, as well as the establishment of new entities through collaboration. Open technological innovation recognizes the essential need of integrating and developing both internal and external resources for new ventures to achieve innovative performance [32].

In essence, open technological innovation is a process in which firms use information technology to rapidly collect, organize, and mine novel internal and external data and knowledge, laying the foundations for technical innovation [33]. The function of IT capability in the open technological creation of new ventures can be described as follows in terms of IT flexibility

and integration: (1) Open technology innovation begins with new ideas, and the breadth and depth of knowledge acquired by new ventures shows the quality of IT capability. The greater a new venture's IT capability, the easier it will be for it to gain internal knowledge and recruit new employees. Knowledge sharing and communication have benefited in the spread of new ideas, thereby contributing in open technological innovation [34]; (2) Recognizing external market need, technical change, and opportunity is a key motivator for open technological innovation. The growth of new ventures' IT capability may help them broaden and deepen their exchanges with their partners, ensuring that new ventures maintain close connections and information sharing with their consumers, suppliers, and partners [35]. This capacity allows firms to quickly identify and integrate new knowledge, technologies, products, and market data, as well as improve corporate management models and promote the development of new products and services. Therefore, we propose the following hypotheses:

*H2*. IT capability positively affects open technological innovation.

*H2a*. IT flexibility positively affects open technological innovation.

*H2b*. IT integration positively affects open technological innovation.

## 2.3 Mediating effect of open technological innovation

Open technological innovation is a model of technological innovation, which is a technological innovation formulated and implemented with an open mind, that is, to acquire and utilize external innovation resources, and to establish and maintain a company's competitive advantage through the integration and utilization of internal and external innovation resources. Open thinking runs through the whole process of technology acquisition, technology management and technology application.

The innovation of new ventures is a very difficult process and it is difficult for entrepreneurs to achieve entrepreneurial success only by relying on their own resources [36]. Enterprises should actively carry out external knowledge search, rationally use and integrate external knowledge, in order to obtain sustainable competitive advantages. The utilization of external knowledge is an important way to improve innovation performance and can more effectively promote the acquisition of complementary resources [37]. The external knowledge search is a trial-and-error procedure that involves continuous exploration and identification of the most effective and valuable methods. Firms must have the ability to absorb external information and implement various management approaches for absorbing imported knowledge. External knowledge must be efficiently integrated when it is presented, which places great demands on the knowledge sharing integration process in new ventures [38]. New firms must have adaptable and integrated information technology capabilities. Moreover, because open innovation focuses obtaining information from the outside and promoting knowledge sharing, there is a possibility of knowledge and technology leakage, which impairs innovation performance acquisition.

The role of open technological innovation has emerged as a central construct in a wide range of organizational studies, including the relationship between open technological innovation and performance [39, 40]. Empirical studies have found a positive relationship between open technological innovation and firm growth [41, 42]. Open technological innovation plays a crucial role in organizational performance. Furthermore, Some scholars believed that IT capability itself did not have a direct effect on firm performance, but must indirectly affect firm performance through some intermediary variables [27]. Liu et al. (2013) have studied the roles of absorptive capacity and supply chain agility between IT capability and firm performance from the perspective of dynamic capabilities. They concluded that IT capabilities do influence dynamic capabilities (including absorptive capacity and supply chain agility), and the dynamic capabilities, in turn, promote firm performance [43].

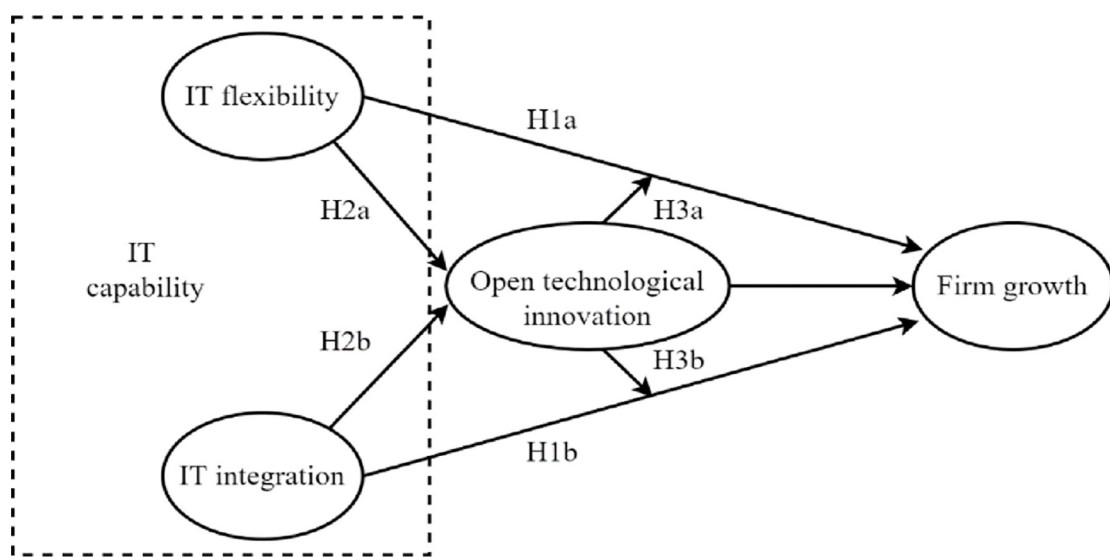

**Fig 1. Theoretical framework and conceptual model.**

According to the intermediary effect test procedure proposed by Baron and Kenny (1986), this study argues that IT capability can improve firm performance by improving the level of open technological innovation [44]. Specifically, with the rapid penetration of information technology in all aspects of enterprise life, enterprises gradually obtain the technical basis for realizing open technological innovation [45]. Enterprises apply information technology capability to strengthen organizational ties and authorization through IT flexibility and IT integration, which promotes open technological innovation within organizations, thus expanding organizational boundaries, and improving organizational efficiency, and ultimately promoting firm growth. Therefore, this study argues that IT flexibility and IT integration can indirectly affect the growth of new ventures by promoting open technological innovation. Based on the above analysis, we propose the following hypotheses:

*H3*. Open technological innovation positively mediates the relationship between IT capability and firm growth.

*H3a*. Open technological innovation positively mediates the relationship between IT integration and firm growth.

*H3b*. Open technological innovation positively mediates the relationship between IT flexibility and firm growth.

From the discussion above, we develop a framework and conceptual model as shown in Fig 1.

## 3. Data and methods

### 3.1 Sample and data collection

The data for this study was obtained from the *China Small and Medium Enterprise Survey* (*CSMES*), which surveyed approximately 10,000 small and medium-sized enterprises (SMEs) in East and North China (Beijing, Tianjin, Shandong, Jiangsu, and Hebei) from January to June 2017. The survey was designed to analyze the environment and development status of SMEs in China and utilized a convenience sampling approach. The collection method employed a mixed system, which involved internet collection in Shandong Province through the Shandong Small and Medium-sized Enterprise Association and mail collection outside of Shandong Province with support from online interviewers. To enhance the reliability of the

**Table 1. Descriptive statistics for sample.**

| Index | Classification | Number | Percentage |
|---|---|---|---|
| Industry | Technology firm | 88 | 34.4% |
| | Non-technical firm | 168 | 65.6% |
| Firm Age | ≤12 months | 18 | 7.0% |
| | 13–24 months | 82 | 32.0% |
| | 25–36 months | 99 | 38.7% |
| | 37–42 months | 57 | 22.3% |

Note: n = 256

survey, the CSMES questionnaires were primarily completed by the CEO, CFO, or R&D department manager. These individuals possess the most critical information on knowledge management and innovation processes within a firm and can provide unique insights into key relevant issues. Additionally, this practice helps to reduce the risk of common method bias.

In this study, a total of 4,168 questionnaires were retrieved, with a recovery rate of 41.7%. There were 2874 valid data values in total, accounting for 68.95% of the overall sample recovery size. Due to the general Chinese government's recent emphasis on "innovation" and "ventures," this research focuses primarily on the research of Chinese new ventures. As a result, we selected 256 valid new venture data from 2,874 valid surveys, based on the definition of new ventures in the global entrepreneurship observation (GEM) report (which refers to firms established within 42 months). The description of the samples is reported in Table 1.

Based on the demographic data, approximately one-third (34.4%) of the new ventures included in this study were technology firms, while the remaining two-thirds (65.6%) were non-technical firms. The data also revealed that the majority of the new ventures (77.7%) were less than 36 months.

## 3.2 Variables and measures

In this study, we utilized established maturity scales to measure variables wherever possible, and made necessary modifications in alignment with the research objectives and context. We employed a Likert-type seven-point scale ranging from "1" (strongly agree/low) to "7" (strongly agree/high) to measure questionnaire items. The variables measured in this study include IT capability, open technological innovation, and firm growth.

**Dependent and independent variables.** *IT capability (ITC).* Previous research has highlighted the importance of IT capability in driving firm performance and innovation. For instance, Rai and Tang (2010) argued that IT capability is a crucial determinant of firm innovation and competitiveness [30]. Similarly, Cui et al. (2015) found that IT capability positively influences firm innovation and financial performance [28]. Bi et al. (2019) also identified IT capability as a key driver of firm innovation and suggested that firms with higher IT capability are better equipped to respond to environmental changes and seize new opportunities [46].

Building upon this literature, our study conceptualizes IT capability as a multidimensional construct that comprises IT flexibility and IT integration. IT flexibility refers to the ability of an organization to adapt and respond to changing business requirements and technological developments through the effective use of IT resources. IT integration, on the other hand, reflects the degree to which an organization integrates its IT systems and processes with other functional areas and external stakeholders.

IT flexibility (*ITF*) was measured by:

1. "The information system can rapidly change and adapt to new conditions."

2. "The information system is highly scalable."

3. "The information system design supports new cooperative innovation relationships."

4. "The information system can be easily combined with new applications and functions."

IT integration (*ITI*) was measured by:

1. "The information system can easily access the information regarding innovative partners."

2. "The resources and information systems can be seamlessly integrated with those of the innovative partners."

3. "The information system can exchange data with innovation partners in real time."

4. "The information systems can easily aggregate databases of innovative partners."

*Open technological innovation (OTI).* In order to gain a competitive edge, enterprises often seek to acquire external knowledge and expertise. The external linkages and sources of knowledge are recognized as critical determinants of innovation. Within the typology of technological innovation strategies, open innovation is comprised of product-oriented and process-oriented approaches [47]. To capture the concept of open technological innovation in our study, we adapted a construct of open technological innovation behavior measured by four items from previous studies [47, 48].

Open technological innovation (*OTI*) was measured by:

1. "New product development can meet expectations."

2. "New product/new service introduction rate is improved."

3. "New method/new service adoption rate is rapidly increasing."

4. "Innovation is more open than competitors."

*Firm growth (FG).* The research objects of this study were Chinese new ventures. For new ventures, future competitive advantage is often best reflected through growth performance rather than profitability. To measure growth performance in our study, we adopted the approach used by Cavazos et al. (2012) and used product market share as a proxy variable [49]. This choice is consistent with prior research, which suggests that market share can be a useful indicator of a firm's growth potential and long-term competitiveness.

**Control variables.** To account for the potential impact of extraneous variables on new venture growth, we included industry type (*IND*), firm age (*FA*), and firm size (*FS*) as control variables in our analysis. This approach is consistent with previous research and enables us to isolate the effects of the primary variables of interest [50]. Industry type was measured using a dummy variable based on the "Guides for High-Tech Enterprise Certification" and used to control for the effects of industry-specific factors on new venture growth. Firm age, on the other hand, was measured as the duration from the establishment of the firm until the end of 2016, and was used to control for the potential influence of age-related factors on growth performance. Lastly, we included firm size as a control variable by measuring it as the logarithm of the sales revenue of firms. This approach is consistent with prior research and allows for a more accurate assessment of the effects of firm size on new venture growth, as it accounts for potential nonlinearities in the relationship between size and growth.

## 3.3 Common method bias

Scholars asserted the influence of common method bias (CMB) in self-reporting variables, which necessitates the measurement of CMB as suggested by experts [51]. Literature has shown several statistical methods to recognize and control for any possible CMB [52]. To avoid the probably existing CMB issues, this study used Harman's single-factor test as suggested by Podsakoff and Organ (1986) to check for the presence of CMB [53]. We used exploratory factor analysis, and Principle Component Factor Analysis has been used to analyze all questionnaire items for constructs with a CMS eigenvalue larger than 1, the cumulative percent of variance is 63.973%. The explanation variance percentage of the first factor is 41.458% (less than 50%). The results show that there is no single factor that explains the majority of the variance across a wide range of data sets, and the first important factor does not explain the majority of the variance. There results suggest that CMB is not a significant issue in our data.

## 3.4 Data analysis methods

Analysis of moment structures (AMOS) was used to measure validation and test the structural model based on the data set of 256 Chinese new ventures. Confirmatory factor analysis (CFA) was implemented to examine the validity and reliability of the constructs. Data analysis was conducted using SPSS version 26 and AMOS version 17. In addition, stepwise regression was used to test hypotheses, which is suitable in this context. Our main objective was to examine associations and relationships among these variables. By utilizing regression analysis, we were able to assess the strength and direction of these relationships while controlling for potential confounding factors. This analytical approach allowed us to generate meaningful insights into the observed associations in the data. [54, 55].

# 4. Empirical results

## 4.1 Reliability and validity

To ensure the proposed model's quality, it must be both statistically reliable and valid. Reliability means producing similar results across multiple tests under the same conditions. Cronbach's alpha is a widely used measure of reliability, and in this study, three measures were examined. The obtained values of 0.826, 0.784, and 0.780, respectively, all higher than 0.75, indicating high reliability of the proposed model.

The validity of an analysis refers to the degree to which it accurately represents the intended information. Validity can be analyzed through content and construct validity analysis methods. Content validity evaluates the extent to which the questionnaire items accurately reflect the intended content. In this study, the questionnaire items were primarily drawn from the maturity scale, which had been reviewed and revised by the research team's professor. During the pilot test, no reports of misunderstandings were received, and interviewees indicated that the items were easy to understand, indicating satisfactory content validity [56].

The confirmatory factor analysis (CFA) is a powerful method used to assess construct validity. According to Campbell and Fiske's seminal work (1959), construct validity research typically examines the degree to which data provide, including convergent validity and discriminant validity [57].

Table 2 presents the results of the CFA for assessing convergent validity. All factor loadings were significant and exceeded the recommended threshold of 0.5, indicating strong convergent validity [58]. To assess the model fit, several well-established goodness-of-fit indices were utilized, including the normed chi-square ($\chi2/df$), the goodness-of-fit index (GFI), the adjusted goodness-of-fit index (AGFI), and the root mean square error of approximation

Table 2. Convergent validity of the measurement model.

| Variable | AVE | CR |
|---|---|---|
| IT flexibility (ITF) | 0.560 | 0.835 |
| IT integration (ITI) | 0.532 | 0.771 |
| Open technological innovation (OTI) | 0.557 | 0.789 |

Notes: AVE is the average variance extracted; CR is the critical ratio; goodness-of-fit indices and recommended thresholds $\chi^2$/df = 2.768 (<3); GFI>0.9; AGFI>0.9; RMSEA<0.05

(RMSEA) [59]. In the measurement model, all indices exceeded their recommended thresholds, indicating sufficient construct validity.

Discriminant validity is demonstrated when the square root of the average variance extracted (AVE) for a construct is greater than the corresponding inter-construct correlations. As Table 3 indicates, the square root of the AVE for each variable exceeded the intercorrelations, indicating sufficient discriminant validity.

## 4.2 Hypothesis testing

**Summary statistics and correlations.** Table 4 shows the descriptive statistics of the main variables. On average, the mean values of ITF, ITI, and OTI are 4.637, 4.358, and 4665, which indicate that most respondents believe that IT flexibility, IT integrality, and open technological innovation are of great significance for the growth of new ventures. The firm age in our sample is around 27 months, with a relatively high standard deviation (SD = 9.561).

Table 4 also describes the correlation coefficients among the main variables. The relationships among IT capability (both IT flexibility and IT integration), open technological innovation and firm growth are positive, thus indicate that IT capability and open technological innovation may help firms achieve higher product market share. Meanwhile, the relationships between IT capability (both IT flexibility and IT integration) and open technological innovation are also positive. This observation is consistent with our hypothesis. The correlation coefficient reflects only a simple correlation between variables. Therefore, further analysis using a regression model is necessary.

**Regression analysis.** *The effects of IT capability and open technological innovation on firm growth*. We employed SPSS 26.0 to do the regression analysis. Table 5 shows the empirical results for the model estimations of the effects of IT capability and open technological innovation on firm growth.

Models 1, 2, and 3 examine whether or not IT capability (both IT flexibility and IT integration) has positive effects on firm growth. Both the coefficients of IT flexibility ($\beta$ = 0.234, $p < 0.01$) and IT integration ($\beta$ = 0.310, $p < 0.01$) are significantly positive, which suggests that a high level of IT capability can improve firm growth. Model 4 examines the effects of open technological innovation on firm growth. The result shows that open technological innovation is

Table 3. Discriminant validity of the measurement model.

| Variable | ITF | ITI | OTI |
|---|---|---|---|
| IT flexibility (ITF) | 0.749 | | |
| IT integration (ITI) | 0.528 | 0.729 | |
| Open technological innovation (OTI) | 0.446 | 0.416 | 0.746 |

Note: The square root of the AVE as a criteria

**Table 4. Descriptive statistics and correlation matrix for main variables.**

| Variable | Mean | SD | 1 | 2 | 3 | 4 | 5 | 6 |
|---|---|---|---|---|---|---|---|---|
| 1. ITF | 4.637 | 1.984 | 1 | | | | | |
| 2. ITI | 4.358 | 1.806 | 0.619*** | 1 | | | | |
| OTI | 4.665 | 1.574 | 0.444*** | 0.412*** | 1 | | | |
| 4. FG | 4.390 | 1.292 | 0.252*** | 0.276*** | 0.369*** | 1 | | |
| 5. IND | 0.340 | 0.476 | 0.124** | 0.079 | 0.145** | 0.035 | 1 | |
| 6. FA | 27.180 | 9.561 | -0.080 | -0.062 | -0.134** | -0.099 | -0.061 | 1 |
| 7. FS | 3.209 | 0.432 | -0.020 | -0.018 | -0.084 | -0.052 | -0.074 | 0.975*** |

Notes

* $p < 0.10$

** $p < 0.05$

*** $p < 0.01$.

beneficial to firm growth, as demonstrated in previous studies. Therefore, these results support *H1*.

*The effects of IT capability on open technological innovation*. Table 6 shows the empirical results for the model estimations of the effects of IT capability on open technological innovation. Models 5, 6, 7 and 8 examine whether or not IT capability (both IT flexibility and IT integration) has positive effects on open technological innovation. Both the coefficients of IT flexibility ($\beta = 0.422$, $p < 0.01$) and IT integration ($\beta = 0.393$, $p < 0.01$) are significantly positive, which suggests that a high level of IT capability can improve open technological innovation, which supports *H2*.

*The mediating effect of open technological innovation*. Based on Baron and Kenny' (1986) three steps for confirming the mediation role, the research should meet three requirements [44]:

1. IT capability: IT flexibility as well as IT integrality should pose significance on firm growth.

2. IT capability: IT flexibility as well as IT integrality should pose significance on the open technological innovation.

**Table 5. Multiple regression analysis of the effects of IT capability and open technological innovation on firm growth.**

| Dependent variables and models | Firm Growth Model 1 | Firm Growth Model 2 | Firm Growth Model 3 | Firm Growth Model 4 |
|---|---|---|---|---|
| IND | 0.030(0.166) | 0.001(0.163) | 0.034(0.162) | -0.019(0.158) |
| FA | -0.189*(0.014) | -0.142(0.014) | -0.021(0.014) | -0.106(0.014) |
| FS | 0.115(0.397) | 0.079(0.388) | 0.348(0.386) | 0.067(0.374) |
| ITF | | 0.234***(0.078) | | |
| ITI | | | 0.310***(0.079) | |
| OTI | | | | 0.349***(0.077) |
| Constant | 3.682***(0.989) | 3.970***(0.967) | 3.829***(0.963) | 3.993***(0.932) |
| Observations | 256 | 256 | 256 | 256 |
| Adjusted $R^2$ | 0.003 | 0.053 | 0.057 | 0.107 |
| F value | 1.248(1.260) | 4.576***(1.228) | 5.595***(1.218) | 9.482***(1.185) |

Notes

* $p < 0.10$

** $p < 0.05$

*** $p < 0.01$.

**Table 6. Multiple regression analysis of the effects of IT capability on open technological innovation.**

| Dependent variables and models | OTI Model 5 | OTI Model 6 | OTI Model 7 | OTI Model 8 |
|---|---|---|---|---|
| IND | 0.139**(0.128) | 0.088(0.117) | 0.109*(0.118) | 0.087(0.115) |
| FA | -0.238**(0.011) | -0.153(0.010) | -0.179*(0.010) | -0.148(0.010) |
| FS | 0.138(0.305) | 0.072(0.278) | 0.094(0.281) | 0.068(0.273) |
| ITF | | 0.422***(0.056) | | 0.287***(0.069) |
| ITI | | | 0.393***(0.056) | 0.219***(0.069) |
| Constant | -0.697(0.760) | -0.291(0.692) | -0.434(0.699) | -0.274(0.680) |
| Observations | 256 | 256 | 256 | 256 |
| Adjusted $R^2$ | 0.032 | 0.204 | 0.183 | 0.231 |
| F value | 3.780***(0.968) | 17.309***(0.878) | 15.275***(0.890) | 16.297***(0.863) |

Notes

* $p < 0.10$

** $p < 0.05$

*** $p < 0.01$.

3. When the open technological innovation is added to the models of IT capability and firm growth, respectively, the standardized estimates of the path of IT capability to firm growth may become insignificant (whole mediation), and may weaken before adding the open technological innovation (part of mediation).

At the same time, it must be noted that open technological innovation has a significant impact on firm growth. H1, H1a, H1b, H2, H2a, and H2b have satisfied the requirements of (1) and (2). Thus, we constructed the mediation models below to confirm the mediation role of open technological innovation.

First, we confirmed the mediation role of open technological innovation between IT flexibility and firm growth; the research model and results can be seen in Table 7 and Fig 2. The standardized estimate of the path of IT flexibility to open technological innovation is 0.422 under the significance level of $p < 0.01$. The standardized estimate of the path of open technological innovation to firm growth is 0.304 and its significance level is $p < 0.001$. The

**Table 7. The mediating effects of open technological innovation.**

| Dependent variables and models | Firm Growth Model 1 | Firm Growth Model 2 | Firm Growth Model 3 | Firm Growth Model 9 | Firm Growth Model 10 |
|---|---|---|---|---|---|
| IND | 0.030(0.166) | 0.001(0.163) | 0.034(0.162) | -0.025(0.158) | -0.022(0.157) |
| FA | -0.189*(0.014) | -0.142(0.014) | -0.021(0.014) | -0.096(0.014) | -0.098(0.013) |
| FS | 0.115(0.397) | 0.079(0.388) | 0.348(0.386) | 0.057(0.374) | 0.059(0.371) |
| ITF | | 0.234***(0.078) | | 0.106*(0.083) | |
| ITI | | | 0.310**(0.079) | | 0.147**(0.081) |
| OTI | | | | 0.304**(0.085) | 0.289***(0.083) |
| Constant | 3.682***(0.989) | 3.970***(0.967) | 3.829***(0.963) | 4.084***(0.931) | 4.066***(0.925) |
| Observations | 256 | 256 | 256 | 256 | 256 |
| Adjusted $R^2$ | 0.003 | 0.053 | 0.057 | 0.123 | 0.132 |
| F value | 1.248(1.260) | 4.576***(1.228) | 5.595***(1.218) | 8.155***(1.181) | 8.770***(0.175) |

Notes

* $p < 0.10$

** $p < 0.05$

*** $p < 0.01$.

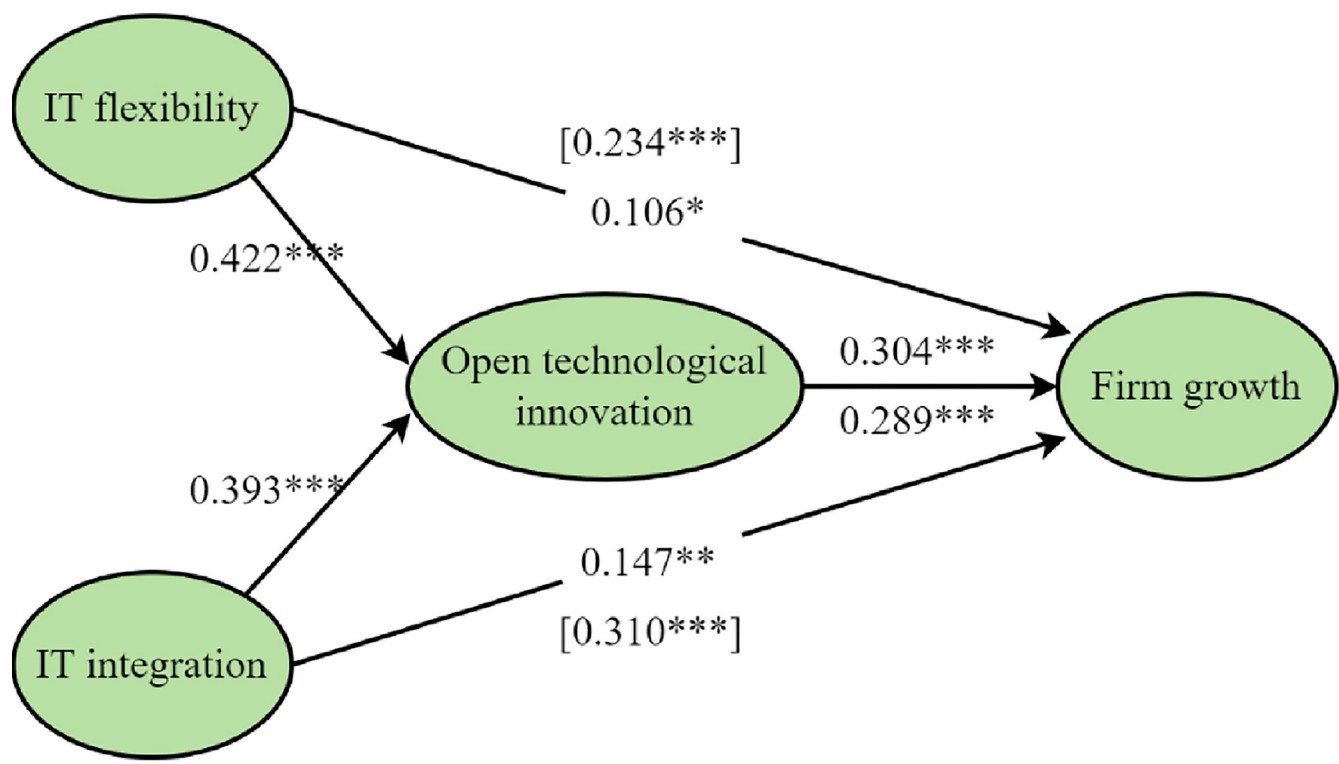

**Fig 2. Path coefficients of the structural model I.**

standardized estimate of the path of IT flexibility to firm growth is 0.106, which is significant under the condition of p<0.1 and lower than the standardized estimate without open technological innovation. The mediating effect accounts for 54.76% of the total effect. The results thus support *H3a*.

Second, we confirmed the meditating role of open technological innovation between IT integrality and firm growth; the research model and results can be seen in Table 5 and Fig 2. The standardized estimate of the path of IT integrality to open technological innovation is 0.393 under the significance level of p<0.01. The standardized estimate of the path of open technological innovation to firm growth is 0.289 and its significance level is p<0.001. The standardized estimate of the path of IT integrality to firm growth is 0.147, which is significant under the condition of p<0.05 and lower than the standardized estimate without open technological innovation. The mediating effect accounts for 43.59% of the total effect. The results thus support *H3b*.

Lastly, we confirmed the mediating role of the open technological innovation effect between IT capability and firm growth; the research model and results can be seen in Fig 3. The standardized estimate of the path of IT capability to open technological innovation is 0.455 under the significance level of p<0.01. The standardized estimate of the path of open technological innovation to firm growth is 0.280 and its significance level is p<0.001. The standardized estimate of the path of IT capability to firm growth is 0.149, which is significant under the condition of p<0.05 and lower than the standardized estimate without open technological innovation. The results show that open technological innovation plays a part of the mediating role between IT capability and firm growth, thus proving *H3*.

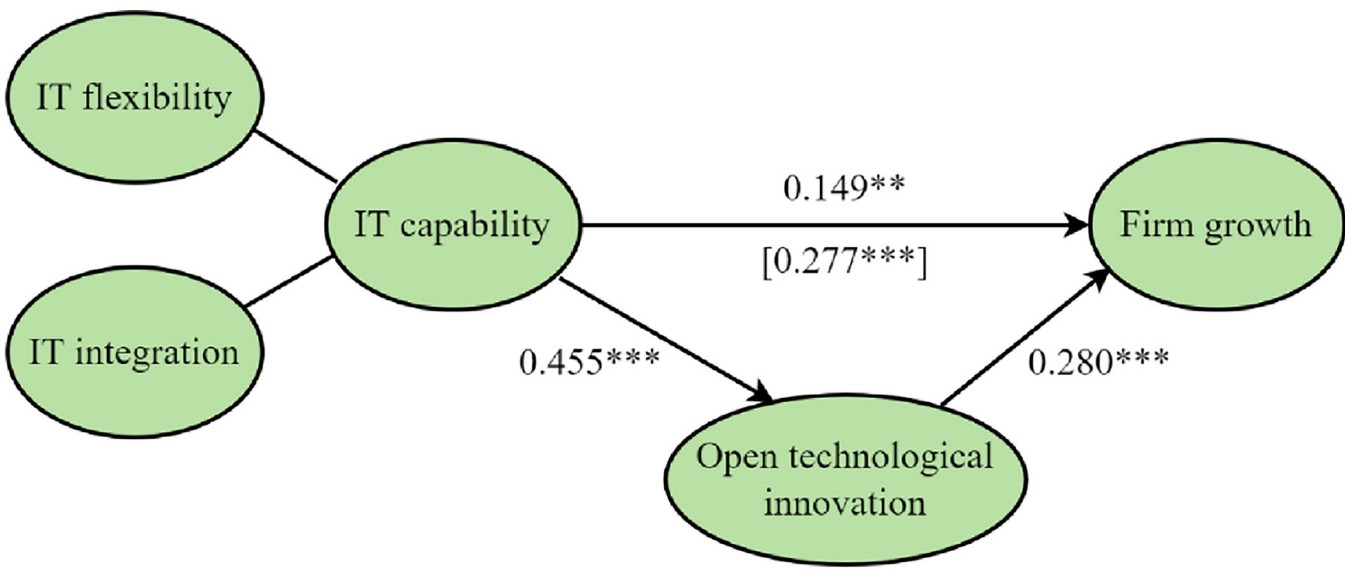

**Fig 3. Path coefficients of the structural model II.**

## 5. Conclusions and discussion

### 5.1 Conclusions

This study contributes to the existing literature on the relationship between IT capability and firm growth by examining the mediating effects of open technological innovation. Our findings are consistent with previous research that suggests IT capability positively influences firm growth [60, 61]. However, our study provides a more nuanced understanding of this relationship by showing that open technological innovation plays a partial mediating role in this relationship. Specifically, our results suggest that new ventures that are able to leverage open technological innovation can better translate their IT capabilities, such as IT flexibility and IT integrality, into firm growth.

This study also extends the literature on open innovation by highlighting its importance in the context of new ventures. Our findings suggest that open technological innovation can be a valuable mechanism for new ventures to access external knowledge, resources, and networks, which can help them overcome resource constraints and compete with established firms [62]. Furthermore, our study shows that open technological innovation can help new ventures to translate their IT capabilities into firm growth. This highlights the importance of managing innovation holistically and considering the interactions between different types of innovation.

### 5.2 Theoretical contribution

The sharing economy has brought about significant changes in the resource connection mode, connection cost, and connection efficiency, enabling new ventures to search for low-cost external resources to replace their internal functions. While the literature has discussed the importance of the sharing economy in solving the problem of misallocation of innovation resources, there is a lack of theoretical basis for new ventures on how to leverage their IT capability to promote firm growth in this context.

This study addresses this gap by providing novel insights into the relationship between IT capability, open technological innovation, and firm growth. Specifically, our study contributes to the literature in three ways. First, we expand on previous research by examining the effects

of IT flexibility and IT integrality on firm growth, thus expanding the theory on the effects of IT capability on firm growth. Second, we provide a specific channel for understanding the underlying mechanism of how IT capability and open technological innovation enhance firm growth, which has not been integrated into empirical research in the literature. Third, our study fills a gap in the literature by examining the effects of innovation management outcomes in emerging economies such as China, where the focus has been on adopting innovation management, which is still largely unexplored.

Overall, this study offers valuable contributions to the literature on innovation management by providing empirical evidence of the mediating effects of open technological innovation in the relationship between IT capability and firm growth. Our findings have important implications for researchers and practitioners interested in enhancing innovation management and improving the competitiveness of new ventures in the context of the sharing economy.

## 5.3 Managerial implications

Our study provides valuable insights into the role of IT capability and open technological innovation in promoting firm growth for new ventures in China. Based on our findings, we offer three important managerial implications for new ventures seeking to expand their competitiveness.

First, to overcome the "newness" problem and obtain rapid growth opportunities, new ventures must creatively utilize their internal resources and develop informal network relations, while also actively adopting the open technological innovation model. This involves strengthening the acquisition of information and technical resources in the external market to make up for the lack of internal innovation resources, as well as promoting the internal technology of the organization to achieve external marketization. These strategies can improve the efficiency of open technological innovation and increase the market competitiveness of new ventures. Furthermore, in the open innovation mode, the boundary of the enterprise becomes fuzzy, and technological innovation has evolved from a "closed" to an open global innovation mode. The open technological innovation system needs to absorb more innovation stakeholders, such as enterprise employees, leading users, suppliers, and technical partners, to form a multi-agent innovation model. Through their interaction, this model can better carry out innovation activities, improve innovation quality, and enhance the effectiveness of innovation management. Overall, adopting a strategic approach to open technological innovation can help new ventures overcome resource constraints and improve their competitiveness in the market.

Second, the abundance of information in the current business environment highlights the importance of new ventures acquiring and effectively utilizing information technology (IT). However, the resource-based theory suggests that only scarce, valuable, difficult-to-imitate, and irreplaceable enterprise resources and capabilities can provide a sustained competitive advantage. IT resources, on the other hand, have become ubiquitous and therefore, may not necessarily provide substantial competitive advantages for new ventures. To achieve a competitive advantage, new ventures need to transform their IT investment into heterogeneous IT capability. This involves developing IT capabilities that are unique to the organization and that cannot be easily replicated by competitors. By doing so, new ventures can realize the strategic value of IT and gain a sustained competitive advantage in the market. Thus, it is crucial for new ventures to focus on developing IT capabilities that are aligned with their business strategy and that can effectively support their innovation activities and growth.

Third, the influence of information technology capability on open technological innovation and firm growth is a systematic and long-term process. To fully utilize IT flexibility and IT

integration, new ventures need to build a comprehensive Internet-information system. Firstly, IT flexibility requires the Internet-information system to be flexible in processing information according to different situations, and to be highly scalable. Additionally, to strengthen knowledge sharing and collaboration capabilities with innovative partners, the system should facilitate the establishment of new collaborative innovation relationships and the integration of new applications and functions. Secondly, IT integration requires the Internet-information system to seamlessly integrate with the system of innovation partners and provide easy access to exchange and aggregate the resources and data of innovation partners. This integration and absorption of resources can help new ventures to improve their innovation capabilities and enhance their market competitiveness. Therefore, new ventures need to prioritize the development of a robust Internet-information system to effectively leverage their IT capability for open technological innovation and firm growth. This involves building collaborative relationships with innovative partners, promoting knowledge sharing and collaboration, and enhancing the integration of resources and data across the innovation ecosystem. By doing so, new ventures can effectively translate their IT capability into competitive advantage and promote their growth in the market.

### 5.4 Limitations and future research

While this study provides important contributions to the literature on the relationship between IT capability, open technological innovation, and firm growth, there are several limitations that need to be addressed in future research. Firstly, the cross-sectional design adopted in this study limits the ability to establish causality between the constructs over time. Future research should use a longitudinal study to overcome this limitation and consolidate the results. Secondly, the smaller sample size limits the generalizability of our findings. The data used in this study may not be representative of all new ventures and may be limited by the interviewees' cognition. To address this limitation, future research should focus on a specific industry in China and explore the effects of IT capability on open technological innovation and firm growth under the regulatory impact of the sharing economy environment to better understand the relationship among them.

Additionally, there may be other variables, such as organizational learning, that mediate the relationship between IT capability and firm growth. Therefore, future research should investigate the role of other mediators to provide a more comprehensive understanding of the mechanisms through which IT capability influences firm growth. By addressing these limitations, future research can further enhance our understanding of the complex relationship between IT capability, open technological innovation, and firm growth in the context of new ventures in China.

## Supporting information

**S1 Data.**
(DTA)

## Author Contributions

**Conceptualization:** Weizhi Yao.

**Data curation:** Weizhi Yao.

**Formal analysis:** Weizhi Yao, Lianshui Li.

**Funding acquisition:** Weizhi Yao, Lianshui Li.

**Investigation:** Weizhi Yao.

**Methodology:** Weizhi Yao, Lianshui Li.

**Project administration:** Weizhi Yao, Lianshui Li.

**Resources:** Weizhi Yao.

**Software:** Weizhi Yao.

**Supervision:** Weizhi Yao, Lianshui Li.

**Validation:** Weizhi Yao, Lianshui Li.

**Visualization:** Weizhi Yao.

**Writing – original draft:** Weizhi Yao.

**Writing – review & editing:** Weizhi Yao, Lianshui Li.

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
