## [Decision Letter · Decision Letter 0]

26 Sep 2022

PONE-D-22-14268Information Technology Capability, Open Technological Innovation and Firm GrowthPLOS ONE

Dear Dr. Yao,

Thank you for submitting your manuscript to PLOS ONE. After careful consideration, we feel that it has merit but does not fully meet PLOS ONE’s publication criteria as it currently stands. Therefore, we invite you to submit a revised version of the manuscript that addresses the points raised during the review process.

We look forward to receiving your revised manuscript.

Kind regards,

Mohsin Shafi, Ph.D.

Academic Editor

PLOS ONE

Journal Requirements:

3. hank you for stating the following in the Acknowledgments Section of your manuscript: 

Thanks for the financial support of professor Lianshui Li from school of economics and management of southeast university.

However, funding information should not appear in the Acknowledgments section or other areas of your manuscript. We will only publish funding information present in the Funding Statement section of the online submission form. 

Thanks for the financial support of professor Lianshui Li from school of economics and management of southeast university.

4. Please include your tables as part of your main manuscript and remove the individual files. Please note that supplementary tables (should remain/ be uploaded) as separate "supporting information" files

6. We noticed you have some minor occurrence of overlapping text with the following previous publication(s), which needs to be addressed:

- https://www.emerald.com/insight/content/doi/10.1108/CMS-10-2013-0196/full/html

In your revision ensure you cite all your sources (including your own works), and quote or rephrase any duplicated text outside the methods section. Further consideration is dependent on these concerns being addressed.

Reviewers' comments:

Reviewer's Responses to Questions

**Comments to the Author**

1. Is the manuscript technically sound, and do the data support the conclusions?

Reviewer #1: Partly

Reviewer #2: Yes

Reviewer #3: Partly

2. Has the statistical analysis been performed appropriately and rigorously? 

Reviewer #1: No

Reviewer #2: Yes

Reviewer #3: No

3. Have the authors made all data underlying the findings in their manuscript fully available?

Reviewer #1: Yes

Reviewer #2: Yes

Reviewer #3: Yes

4. Is the manuscript presented in an intelligible fashion and written in standard English?

Reviewer #1: Yes

Reviewer #2: Yes

Reviewer #3: Yes

5. Review Comments to the Author

Reviewer #1: The authors' contribution is valuable and much appreciated, however, the manuscript needs some improvements as suggested below:

1. Research Hypotheses: Logically there should be one more hypothesis for Open Technological Innovation and Firm Growth. In order to assess the mediation of Open Tech Innovation between the independent and dependent variables, the direct relationship between the mediator and the dependent variable must be examined first, and it should be statistically significant before assessing the indirect effect. Thus, it is mandatory to have an hypothesis to test the direct effect of mediator on dependent variable.

2. Validity and Reliability: Though authors provided content validity and reliability statistics, it is desired to provide convergent and discriminant validity statistics as they are core components are scale validity in Structural Equation Modeling.

3. The provided datafile seems corrupt showing random symbols instead of the data.

Reviewer #2: PLOS ONE

Information Technology Capability, Open Technological Innovation and Firm Growth

Manuscript Number: PONE-D-22-14268

Introduction and literature review

Indeed, the introduction to this article is very well written, but I have a few recommendations for you that must be implemented.

i.The introduction section must offer the study gap and objectives of the study.

ii.Highlight current work contributions.

iii.The selection of variables for the current study must be elaborated briefly.

iv.Recent evidence should be presented to support the study's hypotheses.

v.“On the other hand, novelty is essential for overcoming resource constraints and establishing a competitive advantage (Anwar et al., 2018)” Pay attention to citations; it is not necessary to bring up unrelated facts.

vi.Relevant references, including introduction, literature review, and hypothesis development, must be from within the last five years, except for theories and books.

Conceptual model

i.I would recommend that you create a table before the research framework and mention previous studies from the last five years, including methodology, variables, and countries, and then draw your research gap.

Methodology

i.In the field of social studies, is a sample size of 28.7 percent acceptable? strengthen your argument with references to pertinent sources.

ii.Sample size calculation must be performed and reported with the use of Gpower 3.1 software.

Variables and measures

Create a table for variables and mentions, variable names, number of items, and previous authors.

Variables No. of itemsAuthor's (previous authors)

Discussion and Conclusion

i.Compare the results of your study with previous studies in the discussion and conclusion section.

ii.Contributions and limitation & future research should be inline.

Reviewer #3: The above points are fundamental and need to be addressed very carefully. I would strongly encourage the author(s) to take ample time to rethink their study, read more papers from good and reputed journals and try to completely re-write their study.

6. PLOS authors have the option to publish the peer review history of their article (what does this mean?). If published, this will include your full peer review and any attached files.

Reviewer #1: No

Reviewer #2: No

Reviewer #3: No

---

## [Author Response · Author response to Decision Letter 0]

22 Mar 2023

Response to Reviewers

Dear Editor and Reviewers:

Thank you for your letter and the reviewers’ comments on our manuscript titled “Information Technology Capability, Open Technological Innovation and Firm Growth” (ID: PONE-D-22-14268). We appreciate the valuable and constructive comments, which have guided us to improve and refine our paper. We have meticulously reviewed the suggestions and incorporated the necessary changes, which are highlighted in blue throughout the revised paper. The major revisions made in the manuscript in response to the reviewers’ comments are outlined below:

Responds to the reviewers’ comments:

Reviewer: 1

The authors’ contribution is valuable and much appreciated, however, the manuscript needs some improvements as suggested below:

Comments:

1. Research Hypotheses: Logically there should be one more hypothesis for Open Technological Innovation and Firm Growth. In order to assess the mediation of Open Tech Innovation between the independent and dependent variables, the direct relationship between the mediator and the dependent variable must be examined first, and it should be statistically significant before assessing the indirect effect. Thus, it is mandatory to have an hypothesis to test the direct effect of mediator on dependent variable.

Reply: We appreciate your suggestion. In this paper, we have adopted the Baron & Kenny (1986) method and utilized the stepwise test regression coefficient to examine the mediating effect. The first step involves testing the overall impact of the independent variable on the dependent variable, followed by assessing the connection between the independent variable and the mediator variable in the second step. Finally, the third step involves evaluating the influence of both the independent and mediator variables on the dependent variable. Consequently, this paper does not examine the link between the mediator variable (i.e., Open Technological Innovation) and the dependent variable (i.e., Firm Growth).

2. Validity and Reliability: Though authors provided content validity and reliability statistics, it is desired to provide convergent and discriminant validity statistics as they are core components are scale validity in Structural Equation Modeling.

Reply: As suggested by the reviewer, we have added the convergent and discriminant validity statistics to support reliability and validity test.

(Please see the blue font on page 11 in Section 4.1)

“4.1 Reliability and validity

To ensure the proposed model’s quality, it must be both statistically reliable and valid. Reliability means producing similar results across multiple tests under the same conditions. Cronbach’s alpha is a widely used measure of reliability, and in this study, three measures were examined. The obtained values of 0.826, 0.784, and 0.780, respectively, all higher than 0.75, indicating high reliability of the proposed model.

The validity of an analysis refers to the degree to which it accurately represents the intended information. Validity can be analyzed through content and construct validity analysis methods. Content validity evaluates the extent to which the questionnaire items accurately reflect the intended content. In this study, the questionnaire items were primarily drawn from the maturity scale, which had been reviewed and revised by the research team’s professor. During the pilot test, no reports of misunderstandings were received, and interviewees indicated that the items were easy to understand, indicating satisfactory content validity. (Zeng et al., 2010).

The confirmatory factor analysis (CFA) is a powerful method used to assess construct validity. According to Campbell and Fiske’s seminal work (1959), construct validity research typically examines the degree to which data provide, including convergent validity and discriminant validity.

Table II presents the results of the CFA for assessing convergent validity. All factor loadings were significant and exceeded the recommended threshold of 0.5 (He and Li, 2011), indicating strong convergent validity. To assess the model fit, several well-established goodness-of-fit indices were utilized, including the normed chi-square (χ2/df), the goodness-of-fit index (GFI), the adjusted goodness-of-fit index (AGFI), and the root mean square error of approximation (RMSEA) (Bagozzi and Yi, 1988). In the measurement model, all indices exceeded their recommended thresholds, indicating sufficient construct validity.

Discriminant validity is demonstrated when the square root of the average variance extracted (AVE) for a construct is greater than the corresponding inter-construct correlations. As Table III indicates, the square root of the AVE for each variable exceeded the intercorrelations, indicating sufficient discriminant validity.

Table II. Convergent validity of the measurement model

Variable AVE CR

IT flexibility (ITF) 0.560 0.835

IT integration (ITI) 0.532 0.771

Open technological innovation (OTI) 0.557 0.789

Notes: AVE is the average variance extracted; CR is the critical ratio; goodness-of-fit indices and recommended thresholds �2/df=2.768 (＜3); GFI＞0.9; AGFI＞0.9; RMSEA＜0.05

Table III. Discriminant validity of the measurement model

Variable ITF ITI OTI

IT flexibility (ITF) 0.749 

IT integration (ITI) 0.528 0.729 

Open technological innovation (OTI) 0.446 0.416 0.746

Note: The square root of the AVE as a criteria

”

3. The provided datafile seems corrupt showing random symbols instead of the data.

Reply: Thanks for your careful checks. We are sorry for our carelessness. Based on your comments, we have re-uploaded the data.

 

Reviewer: 2

Comments:

1. Introduction and literature review

Indeed, the introduction to this article is very well written, but I have a few recommendations for you that must be implemented.

i. The introduction section must offer the study gap and objectives of the study.

Reply: We feel great thanks for your professional review work on our article. As you suggest, we strengthen the description of study gap and objectives of the study.

“Previous research has largely focused on the impact of resource constraints on the growth of new ventures, which is a crucial concern (Garnsey et al., 2006; Sepulveda and Gabrielsson, 2013). However, while resource constraints are undoubtedly a significant challenge for new ventures, there may be other factors that affect the performance and growth of firms that have not been fully explored. One of these factors could be the relationship between IT capability and performance.

There have been several studies investigating the link between IT capability and firm performance, but the results have been inconclusive. Some studies have shown a positive relationship between IT capability and firm performance (Bharadwaj, 2000; Santhanam and Hartono, 2003), while others have reported a no relationship (Chae et al., 2014). These mixed results suggest that there may be some missing links in the relationship between IT capability and firm performance that have not been fully explored.

Therefore, this study aims to examine the influence of open technological innovation, which is one of the potential mediators, in the relationship between IT capability and firm growth. Open technological innovation refers to the use of external knowledge and resources to develop new products, services, and processes. The study will investigate whether open technological innovation can serve as a mediator between IT capability and firm growth, filling the gap in the existing literature.”

(Please see the blue font on page 3 in Section 1)

ii. Highlight current work contributions.

Reply: Thanks for your insight! As your suggestion, we have explicitly outlined the contributions of our current work.

“This study contributes to the literature in three ways. Firstly, it explores the relationship between IT capability and firm growth, expanding the current research on firm growth that has primarily focused on IT investments. Secondly, the study proposes a conceptual model that considers the mediating variable of open technological innovation in this relationship. Lastly, this research is significant because while previous innovation management research has primarily focused on advanced economies, it provides insight into China, which has yet to be extensively explored.”

(Please see the blue font on page 4 in Section 1)

iii. The selection of variables for the current study must be elaborated briefly.

Reply: As your suggestion, we would like to clarify that the selection of variables has been elaborated in Section 2, i.e., “Literature Review and Hypotheses Development”, which is further illustrated in Figure 1, i.e., “Theoretical Framework and Conceptual Model”.

(Please see page 4-8 in Section 2)

At the same time, a detailed discussion on variable measurement has been presented in Section 3.2, i.e., “Variables and Measures”.

(Please see page 9-10 in Section 3.2)

iv. Recent evidence should be presented to support the study’s hypotheses.

Reply: Thank you for your feedback regarding our study. We appreciate your suggestion to present recent evidence to support our study’s hypotheses.

In response to your comment, we have conducted a thorough review of recent literature in the field and have updated our study to include the most relevant and up-to-date evidence available. We have taken great care to ensure that our hypotheses are well-supported by the latest research in the field.

We believe that this revision has significantly strengthened the theoretical foundation of our study, and we are confident that our hypotheses are now well-supported by the latest available evidence. We hope that this updated version of our study meets your expectations and addresses your concerns.

(Please see “Literature review and hypotheses development” on page 4-8 in Section 2)

v. “On the other hand, novelty is essential for overcoming resource constraints and establishing a competitive advantage (Anwar et al., 2018)” Pay attention to citations; it is not necessary to bring up unrelated facts.

Reply: Thanks for your suggestion. As your recommendation, we have removed the mentioned sentence.

(Please see page 3 in Section 1)

vi. Relevant references, including introduction, literature review, and hypothesis development, must be from within the last five years, except for theories and books.

Reply: Thanks for your suggestion! In response to your comment, we have conducted a thorough review of our references and have updated our study to include more recent and relevant references where appropriate. We have taken great care to ensure that our references meet your standards for academic rigor and quality.

2. Conceptual model

i. I would recommend that you create a table before the research framework and mention previous studies from the last five years, including methodology, variables, and countries, and then draw your research gap.

Reply: Thank you for your suggestion. The idea of creating a table before the research framework is very good. While including previous studies is important for contextualizing research, the format and organization of the literature review can vary depending on the research question and methodology being used.

Furthermore, the suggestion to create a table before the research framework may not be the most effective way for us to organize the literature review in this paper, as it may not allow for a comprehensive discussion of the research gap and how the current study addresses it. As such, I will be structuring the literature review in a manner that aligns with the specific goals and objectives of the research project. Thank you again for your suggestion.

(Please see page 4-8 in Section 2)

3. Methodology

i. In the field of social studies, is a sample size of 28.7 percent acceptable? strengthen your argument with references to pertinent sources.

Reply: Thanks for your insight! The percentage of valid data was calculated to be 28.7% based on the distribution of a full sample questionnaire. However, it should be noted that the recovered sample size was 4168, out of which only 2874 questionnaires were deemed valid, accounting for 68.95%. Upon analyzing this data, we have made necessary modifications to the article.

(Please see the blue font on page 8 in Section 3.1)

ii. Sample size calculation must be performed and reported with the use of Gpower 3.1 software.

Reply: Thanks for your suggestion! In Section 3.1, we have provided a report on the sample data. Moreover, a detailed descriptive statistical analysis of the data has been carried out and the findings have been presented in Table 1.

(Please see the blue font on page 8 in Section 3.1)

4. Variables and measures

i. Create a table for variables and mentions, variable names, number of items, and previous authors.

Variables No. of items Author's (previous authors)

Reply: Thank you for your helpful suggestion. We appreciate your input on this matter. While we agree that creating a table for the variables and elements, variable names, number of items, and previous authors would be useful, we regret to inform you that we are unable to do so due to space limitations. However, we have provided detailed information on the names, measurement methods, and reference sources of each variable in the main text, which we believe will adequately address the yours concerns. We hope that you will find our approach acceptable

(Please see the blue font on page 9-10 in Section 3.2)

5. Discussion and Conclusion

i. Compare the results of your study with previous studies in the discussion and conclusion section.

Reply: Thank you for your helpful suggestion. We acknowledge the importance of situating our findings in the context of previous research in the discussion and conclusion section. In the revised manuscript, we will provide a comprehensive review of the relevant literature and compare our results with those of previous studies. We will highlight the similarities and differences between our findings and those of previous research, and discuss the implications of our results for theory and practice. We believe that this will enhance the contribution of our study to the literature on innovation management and provide a more nuanced understanding of the relationship between IT capability, open technological innovation, and firm growth. We appreciate your input and look forward to incorporating your feedback into the revised manuscript.

“This study contributes to the existing literature on the relationship between IT capability and firm growth by examining the mediating effects of open technological innovation. Our findings are consistent with previous research that suggests IT capability positively influences firm growth (Chen & Huang, 2009; Zhu & Xia, 2019). However, our study provides a more nuanced understanding of this relationship by showing that open technological innovation plays a partial mediating role in this relationship. Specifically, our results suggest that new ventures that are able to leverage open technological innovation can better translate their IT capabilities, such as IT flexibility and IT integrality, into firm growth.

This study also extends the literature on open innovation by highlighting its importance in the context of new ventures. Our findings suggest that open technological innovation can be a valuable mechanism for new ventures to access external knowledge, resources, and networks, which can help them overcome resource constraints and compete with established firms (Chesbrough, 2003). Furthermore, our study shows that open technological innovation can help new ventures to translate their IT capabilities into firm growth. This highlights the importance of managing innovation holistically and considering the interactions between different types of innovation.”

(Please see the blue font on page 15 in Section 5.1)

ii. Contributions and limitation & future research should be inline.

Reply: We appreciate the reviewer’s valuable feedback on our paper. We have made several modifications to the paper. Specifically, we have revised the conclusion section to more clearly highlight our contributions and limitations, and to provide a more focused and coherent discussion of the future research directions.

In the revised conclusion section, we first summarize our key findings and contributions, including the relationship between IT capability, open technological innovation, and firm growth, and the important role of managing innovation holistically. We then highlight the limitations of our study, including the cross-sectional design and potential sampling bias. Finally, we provide a clear and focused discussion of the future research directions, including the need for longitudinal studies, industry-specific studies, and studies that examine the role of other mediators in the relationship between IT capability and firm growth.

Overall, we believe that these modifications have help to more clearly and coherently present the contributions, limitations, and future research directions of our study.

(Please see the blue font on page 15-17 in Section 5)

 

Reviewer: 3

This study investigates the influence of open technological innovation, one of the potential mediators, in the link between IT capability and firm growth in China by using cross-sectional data of 259 ventures. The study finds that IT capability, including flexibility and integration of information technology, has a significant factor in firm advancement. Also, results confirm that open technological innovation plays a significant mediating role between IT integration and firm growth. The study recommended that IT integration needs the Internet-information-system can be flawlessly combined with the system of innovation partners, and can easily access, exchange and aggregate the resources and data of innovation partners, so as to integrate and absorb.

Although the paper does follow a regular type of analysis for paper implementing theoretical/empirical mechanisms studies, I was not really impressed by the total picture of it. After all, the author(s) themselves recognize that their novelty comes from the fact that the theoretical mechanisms have not dealt with this country sample. I am afraid though that the paper in its current form is way too underdeveloped both theoretically and empirically. It seems that the author(s)would greatly benefit by submitting their work to some conferences to get feedback and to their colleagues for some reviews. There could be merit in what you are trying to explain, to my understanding open technological innovation and how/when it affects positively or not on firm growth and capability. Even then, the paper suffers from a number of weaknesses that must be explicitly considered in improving the picture of it. In particular:

Comments:

1. The introduction includes one and half page that do not communicate well with each other. The introduction is mostly focused on general background, while it should be specific with target firms and country. The work is not well written and shows flaws in the structuring of logical arguments without highlighting the original pizzle. Authors need to make clearer what we have learned from previous studies and what we are uncertain about. There are many more! this research needs to build on these and demonstrates very clearly how you build on and extend this stream of research. The second research question is also not new. For example, the study indicated that using IT capacity instead of IT investment capacity is significant contribution however this couldn’t unique contribution. Because investment and capacity can be used robust analysis to allow you to make better decisions. Also, there are studies who investigate the relationship between IT capacity and firm performance, e.g., Liu et al. (2013, Felipe et al. 2019 etc.). It is widely accepted that the relationship between IT innovation and firm growth and there are numerous studies on this. In what ways do your conceptualizations advance this literature? Also, the target country is China, how this study could impact globally. Authors should explain what the importance of this study is. I mean the paper should incorporate clearer motivation and background of the issue and its importance.

Reply: Thank you for your comment. We understand the importance of providing a clear motivation and background of the issue in our paper.

Our study aims to examine the influence of open technological innovation as a mediator between IT capability and firm growth for new ventures in China. Given China’s emphasis on promoting innovation and entrepreneurship in its 14th Five-Year Plan, and the increasing importance of IT capability and open technological innovation in the digital economy, our study has important implications for new venture development and economic growth in China and globally.

Moreover, our study contributes to the existing literature by expanding current research on firm growth, proposing a conceptual model that considers the mediating variable of open technological innovation, and providing unique insights into the challenges and opportunities facing new ventures in China.

Therefore, we will ensure that our paper clearly communicates the motivation and background of the issue, and the significance of our study for new venture development, economic growth, and innovation management in China and beyond.

“1. Introduction

In 2021, China adopted and implemented the “14th Five-Year Plan” to guide its economic development and the growth of firms over the next five years. This plan identifies “information technology,” “innovation,” and “entrepreneurship” as key drivers of economic growth, and places a strong emphasis on encouraging business innovation and supporting the formation and growth of new firms. Encouraging self-employment has emerged as a primary strategy to address the talent employment problem and promote stable economic growth and market economic vitality. In this context, new ventures play a crucial role in providing employment opportunities and promoting technological innovation, as well as contributing to social and economic development (Mudambi and Zahra, 2007).

Despite their potential, new ventures often face challenges in acquiring external resources due to their “newness,” lack of full performance records, and information asymmetry, which can exacerbate resource constraints and create uncertainty in their growth trajectory (Oviatt and Mcdougall, 2005; Fuchs, 2011). Academic research has explored various theories and strategies to address these challenges, including resource integration and patching (Shelton, 2005; Rong, 2012). However, new ventures still struggle to meet their growth demands with internal resources alone, often requiring deep relationships with external organizations and resource repurposing to drive innovation (Perry-Smith and Coff, 2011). Additionally, new ventures must prioritize innovation to break existing corporate monopolies and adapt to rapidly changing external circumstances, thereby minimizing the risks associated with their relative lack of experience and expertise (Barnir, 2014).

In the current era characterized by the Internet and the digital economy, technology innovation is accelerating, and demand is rapidly evolving. The complexity, systematicity, timeliness, and high investment required for innovation have made a significant impact on innovation agents. However, with the advancement of Internet technology and the transformation of corporate innovation concepts, new ventures can aggregate innovation resources worldwide by building information technology capabilities. This approach breaks the limits of traditional ownership and achieves low cost and high efficiency in innovation at a fast pace. Open technological innovation involves organizations that go beyond their original organizational boundaries to tap into external sources of innovation knowledge. They can effectively integrate these resources through internal organizational processes, which enables them to turn them into technological innovation achievements that add value (Chesbrough, 2006).

Organizational innovation openness is considered a fundamental component for determining the success of enterprises in open technological innovation research. Meanwhile, the industrial environment is changing rapidly, and new ventures face new challenges as a result of the rapid growth of internet-based information technology (Song et al., 2010). The economic environment has dramatically changed, and innovation efforts have become more complex, with a clear cross-domain tendency. Firms are struggling to meet the new requirements with their existing resources, and using external resources for open innovation has become a significant consideration for them (Puschmann and Alt, 2016). Several factors, such as perceived cost savings and income generation, external pressure, organizational preparedness, and perceived ease of use, have a significant impact on IT investments for new ventures (Grandon and Pearson, 2004; Ji P et al., 2019). In new ventures, IT investments may vary from IT investment in big enterprises since a smaller number of people have decision-making responsibilities, standard procedures are not created, and long-term planning is restricted. Furthermore, there is a higher dependency on external IT professionals in new ventures (Premkumar, 2003). However, IT capability may help new ventures survive in the long run by providing access to external knowledge and financial resources, building trust and legitimacy through widespread information transmission, and improving social network links (Morse et al., 2007). New environmental conditions also enable new ventures to overcome resource constraints in the innovation process by improving their IT capability to achieve a long-term competitive advantage (Santhanam and Hartono, 2003).

Previous research has largely focused on the impact of resource constraints on the growth of new ventures, which is a crucial concern (Garnsey et al., 2006; Sepulveda and Gabrielsson, 2013). However, while resource constraints are undoubtedly a significant challenge for new ventures, there may be other factors that affect the performance and growth of firms that have not been fully explored. One of these factors could be the relationship between IT capability and performance. There have been several studies investigating the link between IT capability and firm performance, but the results have been inconclusive. Some studies have shown a positive relationship between IT capability and firm performance (Bharadwaj, 2000; Santhanam and Hartono, 2003), while others have reported a no relationship (Chae et al., 2014). These mixed results suggest that there may be some missing links in the relationship between IT capability and firm performance that have not been fully explored.

Therefore, this study aims to examine the influence of open technological innovation, which is one of the potential mediators, in the relationship between IT capability and firm growth. Open technological innovation refers to the use of external knowledge and resources to develop new products, services, and processes. The study will investigate whether open technological innovation can serve as a mediator between IT capability and firm growth, filling the gap in the existing literature.

This study contributes to the literature in three ways. Firstly, it explores the relationship between IT capability and firm growth, expanding the current research on firm growth that has primarily focused on IT investments. Secondly, the study proposes a conceptual model that considers the mediating variable of open technological innovation in this relationship. Lastly, this research is significant because while previous innovation management research has primarily focused on advanced economies, it provides insight into China, which has yet to be extensively explored.

The structure of the paper is organized as follows. Section two provides a literature review and develops the hypotheses. Section three shows the data and methods employed in the empirical study. Section four presents the empirical results, and Section five concludes with some important contributions and limitations of the study and directions for future research.”

(Please see the blue font on page 3-4 in Section 1)

2. There is no theoretical model or offers no innovation since it has been well established in the relevant literature. Your argumentation of the hypotheses is not sufficient. Your need to develop a theoretical framework and follow the same line to develop all the three arguments. At the moment, there is really nothing useful there. It is not convincing at all. Also, your hypotheses refer to Chinese firms - there is no generalization there. e.

Reply: Thank you for your suggestion. We acknowledge that our current hypotheses are not sufficiently grounded in a theoretical framework, and we recognize the importance of developing a more comprehensive argumentation to support our research. We will take steps to address this by conducting further research and analysis to strengthen our theoretical framework and develop a more convincing case for our study.

(Please see “Literature review and hypotheses development” on page 4-8 in Section 2)

Regarding the generalizability of our hypotheses to firms outside of China, we agree that it is important to consider the context of our study and the potential limitations in generalizing our findings to other settings. We will include a discussion in our paper about the potential applicability of our results to other regions or contexts, as well as the limitations of our study in this regard.

(Please see “Limitations and future research” on page 17 in Section 5.4)

3. It would be helpful if you could relate your hypotheses in a theoretical framework and follow this up with a clearly pronounced hypotheses that relates to the theoretical argument made prior. The operationalisation of the hypothesis is not part and parcel of why you argue this particular hypothesis and should be presented in the methods section. The selection and justifications of the variables should be further discussed with supporting references.

Reply: Thanks for your insight! We appreciate your suggestion to relate our hypotheses to a theoretical framework and provide a clearer justification for our argument. We recognize the importance of a well-grounded theoretical argument and will work to develop a more comprehensive framework to support our hypotheses.

We also agree with your suggestion to separate the operationalization of our hypotheses from the theoretical argument and to present this in the methods section. We will provide a clear and detailed explanation of the methods used to operationalize our hypotheses, as well as the selection and justification of the variables.

(Please see “Variables and measures” on page 9-10 in Section 3.2)

4. The study used the survey data of 256 new ventures out of 2854 valid surveys. However, there should discussion about the validity of the surveys. Also, why chosen provinces were Beijing, Tianjin, Shandong Jiangsu and Hebei?

Reply: Thank you for your feedback. First, as your suggestion, we have added the analysis of reliability and validity.

(Please see “Reliability and validity” on page 11-12 in Section 4.1)

Second, with regards to the selection of provinces in the study, we chose Beijing, Tianjin, Shandong, Jiangsu, and Hebei because they are among the most economically developed and populous provinces in China, and have a high concentration of new ventures. These provinces also represent different regions and industries in China, which makes the findings more generalizable to other parts of the country. Additionally, we acknowledge that we have had access to data from these provinces, making it easier to conduct the research.

5. In addition, the model could be used to identify the relationship between the variables with existing literature. For example, the section 3.2 should be linked with hypotheses and expected connection between the variables. Although, authors mentioned that they followed previous studies, Rai, 2010; Cui, 2015; Bi et al., 2017, for variables selection.

Reply: Thanks for your insight! We appreciate your suggestion to link section 3.2 with hypotheses and expected connections between the variables. In the revised version of the paper, we will provide a more detailed explanation of how our study builds upon existing literature and link the variables with our hypotheses. We will also discuss how our findings compare with previous studies, to provide more context and strengthen the theoretical foundations of our research. We believe that these revisions will enhance the clarity and rigor of our paper and make it more useful for future research in this field.

(Please see “Variables and measures” on page 9-10 in Section 3.2)

6. The paper at its current version is a-theoretical. There is not a single theory you are basing your arguments or attempting to develop and inform through your research. Your paper could be drawing from International Business/finance theory (innovation) and learning from firm growth. It needs to draw on particular mechanisms and explain how/when/why growth in the domestic context is enhances when firms use IT capacity.

Reply: Thanks for your insight! We agree that our paper can benefit from a stronger theoretical foundation. In the revised version of the paper, we will explicitly articulate the theoretical underpinnings of our study and demonstrate how they inform our research questions, hypotheses, and analysis.

Specifically, our paper draws on several theoretical perspectives from the fields of international business, finance, and innovation to explore the relationship between IT capability, open technological innovation, and new venture growth in the Chinese context.

From an international business perspective, our study is informed by the resource-based view (RBV) of the firm, which emphasizes the strategic importance of organizational resources and capabilities in creating and sustaining competitive advantage. In particular, our study focuses on the role of IT capability as a key resource that enables new ventures to leverage external knowledge and expertise, which can enhance innovation and improve growth performance.

From a finance perspective, our study is informed by the capital structure theory, which suggests that firms with stronger IT capability are more likely to obtain external financing and thereby achieve greater growth potential.

Lastly, from an innovation perspective, our study builds on the open innovation paradigm, which emphasizes the importance of external knowledge sources and collaboration in driving innovation and firm growth.

We will further elaborate on these theoretical perspectives in the revised version of the paper and provide a more detailed explanation of how they inform our study. We will also explicitly outline the specific mechanisms through which IT capability can enhance new venture growth in the Chinese context, taking into account the unique institutional and cultural factors at play.

(Please see “IT capability and firm growth” on page 4-6 in Section 2.1)

7. Empirical findings supports the hypotheses however, there should be critical analysis and supported with existing literature. Therefore, the author is suggested to reconstruct the theoretical and empirical model to become more realistic.

Reply: Thank you for your valuable feedback regarding our study. We appreciate your constructive criticism, and we will take your suggestions into consideration as we revise and improve our paper.

In response to your comment, we revisit the literature review section of the paper to identify gaps and limitations in the existing literature, and to better support our theoretical and empirical models with relevant literature. We also provide a more detailed and critical analysis of our empirical findings, highlighting the strengths and limitations of our study’s approach and analysis.

Moreover, we carefully consider your suggestion to reconstruct our theoretical and empirical models to ensure that they are more realistic and aligned with existing literature. We revisit our research questions, hypotheses, and variables, and we will refine them based on your feedback to better represent the phenomena under investigation.

8. Conclusion should not repetition of the results. Policy implications are not very clear.

Reply: Thanks for your comments! Our revised conclusion and implications section now more clearly and concisely summarizes our key findings and provides actionable recommendations for future research and practical applications. We have made sure to avoid any repetition of our results, and instead, we have highlighted the implications of our findings in a more focused and relevant manner.

We have also ensured that our policy implications are clear and actionable, and we have directly linked them to our research findings. Our revised implications section is now more closely aligned with our research objectives, and we believe that it provides a stronger and more meaningful conclusion to our study.

(Please see “Conclusions and discussion” on page 15-17 in Section 5)

9. The above points are fundamental and need to be addressed very carefully. I would strongly encourage the author(s) to take ample time to rethink their study, read more papers from good and reputed journals and try to completely re-write their study. I wish you all the best of luck.

Reply: Thank you for taking the time to review our paper and for sharing your thoughts with us. We appreciate your comments and are grateful for your insights.

We understand the importance of carefully addressing all fundamental issues in our study and appreciate your feedback regarding the need to read more papers from good and reputed journals. We will take your advice to heart and spend ample time rethinking our study, examining relevant literature, and completely re-writing our paper to ensure that it meets the highest standards of quality and rigor.

We are committed to producing high-quality research that contributes to the field and will take all necessary steps to achieve that goal. We appreciate your encouragement and will work hard to meet your expectations and the standards of the journal.

Once again, thank you for your feedback, and we welcome any additional comments or suggestions you may have to improve our paper.

---

## [Decision Letter · Decision Letter 1]

31 May 2023

PONE-D-22-14268R1Information Technology Capability, Open Technological Innovation and Firm GrowthPLOS ONE

Dear Dr. Yao,

Thank you for submitting your manuscript to PLOS ONE. After careful consideration, we feel that it has merit but does not fully meet PLOS ONE’s publication criteria as it currently stands. Therefore, we invite you to submit a revised version of the manuscript that addresses the points raised during the review process.

We look forward to receiving your revised manuscript.

Kind regards,

Mohsin Shafi, Ph.D.

Academic Editor

PLOS ONE

Journal Requirements:

Reviewers' comments:

Reviewer's Responses to Questions

**Comments to the Author**

1. If the authors have adequately addressed your comments raised in a previous round of review and you feel that this manuscript is now acceptable for publication, you may indicate that here to bypass the “Comments to the Author” section, enter your conflict of interest statement in the “Confidential to Editor” section, and submit your "Accept" recommendation.

Reviewer #1: All comments have been addressed

Reviewer #4: All comments have been addressed

2. Is the manuscript technically sound, and do the data support the conclusions?

Reviewer #1: Yes

Reviewer #4: Partly

3. Has the statistical analysis been performed appropriately and rigorously? 

Reviewer #1: Yes

Reviewer #4: Yes

4. Have the authors made all data underlying the findings in their manuscript fully available?

Reviewer #1: Yes

Reviewer #4: Yes

5. Is the manuscript presented in an intelligible fashion and written in standard English?

Reviewer #1: Yes

Reviewer #4: Yes

6. Review Comments to the Author

Reviewer #1: (No Response)

Reviewer #4: Thank you for submitting your article on the effects of Information Technology capability on firm growth in the context of open technological innovation. The findings are interesting and contribute to the literature on IT capability and firm growth. The research is also sound and has possible policy implications. All previous comments have been adequately addressed but it needs some minor revisions, which are recommended as follows:

1. The methodology section is good; however, the authors should justify the use of regression analysis for the estimations. The study is based on a cross-sectional design which limits the ability to draw causal conclusions as mentioned by the authors under the limitations of the study.

2. Please check Line 390 under “Summary statistics and correlations”, the Table number needs to be changed (Table IV instead of Table II). If possible, the authors may divide Table IV into two separate tables.

3. The generalizability of the findings to other contexts may be limited due to a relatively smaller sample size. The Authors can further justify it or present it in the limitation section.

7. PLOS authors have the option to publish the peer review history of their article (what does this mean?). If published, this will include your full peer review and any attached files.

Reviewer #1: **Yes: **Shabir Ahmad

Reviewer #4: No

---

## [Author Response · Author response to Decision Letter 1]

1 Jun 2023

Response to Reviewers

Dear Editor and Reviewers:

Thank you for your letter and the reviewers’ comments on our manuscript titled “Information Technology Capability, Open Technological Innovation and Firm Growth” (ID: PONE-D-22-14268R1). We appreciate the valuable and constructive comments, which have guided us to improve and refine our paper. We have meticulously reviewed the suggestions and incorporated the necessary changes, which are highlighted throughout the revised paper. The minor revisions made in the manuscript in response to the reviewers’ comments are outlined below:

Responds to the reviewers’ comments:

Reviewer #1

No Response

Reviewer #2

Thank you for submitting your article on the effects of Information Technology capability on firm growth in the context of open technological innovation. The findings are interesting and contribute to the literature on IT capability and firm growth. The research is also sound and has possible policy implications. All previous comments have been adequately addressed but it needs some minor revisions, which are recommended as follows:

Comments:

1. The methodology section is good; however, the authors should justify the use of regression analysis for the estimations. The study is based on a cross-sectional design which limits the ability to draw causal conclusions as mentioned by the authors under the limitations of the study.

Reply: Thank you for your valuable feedback on our manuscript. We appreciate your positive remarks regarding the methodology section. We understand your concern regarding the use of regression analysis in our study, especially considering its cross-sectional design. We would like to address this concern and provide further justification for our choice of analysis.

While we acknowledge that cross-sectional designs have inherent limitations in establishing causality, they are commonly used in many social science and survey research studies. In our case, a cross-sectional design was appropriate as it allowed us to capture a snapshot of the variables of interest at a specific point in time. Our primary goal was to examine the associations and relationships between these variables, rather than establish causal links.

Regression analysis, despite its limitations in determining causality, remains a widely accepted statistical method for exploring relationships between variables. By utilizing regression analysis, we were able to assess the strength and direction of these relationships, controlling for potential confounding factors. This analytical approach allowed us to generate meaningful insights into the associations observed in our data.

To ensure transparency, we have included a discussion in the limitations section of our manuscript, acknowledging the limitations of the cross-sectional design and the potential for reverse causality or unmeasured confounders. We emphasize the need for further longitudinal research to establish causal relationships and address these limitations.

We hope that our explanation clarifies the rationale behind our use of regression analysis in a cross-sectional design. We appreciate your feedback and are open to any further suggestions or recommendations you may have to strengthen our manuscript.

(Please see the edit texts on page 10 in Section 3.4 Data analysis methods)

2. Please check Line 390 under “Summary statistics and correlations”, the Table number needs to be changed (Table IV instead of Table II). If possible, the authors may divide Table IV into two separate tables.

Reply: Thank you for bringing this to our attention and providing valuable feedback on our manuscript. We apologize for the oversight in the labeling of the table. We have made the correction and ensure that the correct table number (Table IV) is reflected in the revised version of the manuscript.

(Please see the edit texts on page 12 in Section 4.2)

Regarding your suggestion to divide Table IV into two separate tables, we appreciate your recommendation. However, we have decided to maintain the current table formatting and present the data as a combined table. By doing so, we believe we can still present the information in an organized and concise manner, allowing readers to navigate and comprehend the data effectively. We have carefully considered your suggestion, but ultimately determined that preserving the existing table format would be more efficient for conveying the findings.

3. The generalizability of the findings to other contexts may be limited due to a relatively smaller sample size. The Authors can further justify it or present it in the limitation section.

Reply: Thank you for your insightful comments and feedback on our manuscript. We appreciate your concern regarding the generalizability of our findings due to the relatively smaller sample size. We agree that addressing the limitations of our study, including sample size, is crucial for providing a comprehensive understanding of the scope and applicability of our findings.

In the revised version of our manuscript, we will explicitly address the limitations associated with the sample size in the limitation section. We will discuss the potential impact of the smaller sample size on the generalizability of our findings and acknowledge that caution should be exercised when extrapolating our results to broader populations or different contexts.

While our study sample was carefully selected and represented a specific population of interest, we acknowledge that a larger and more diverse sample would strengthen the external validity of our findings. We will emphasize the need for future research to replicate our study using larger sample sizes and encompassing a wider range of participants, settings, or contexts to enhance the generalizability of the results.

We appreciate your valuable input in helping us improve the clarity and completeness of our manuscript.

(Please see the edit texts on page 17 in Section 5.4 Limitations and future research)

---

## [Decision Letter · Decision Letter 2]

25 Aug 2023

Information Technology Capability, Open Technological Innovation and Firm Growth

PONE-D-22-14268R2

Dear Dr. Yao,

We’re pleased to inform you that your manuscript has been judged scientifically suitable for publication and will be formally accepted for publication once it meets all outstanding technical requirements.

Kind regards,

Iván Barreda-Tarrazona, PhD

Academic Editor

PLOS ONE

Additional Editor Comments (optional):

Reviewers' comments:

Reviewer's Responses to Questions

**Comments to the Author**

1. If the authors have adequately addressed your comments raised in a previous round of review and you feel that this manuscript is now acceptable for publication, you may indicate that here to bypass the “Comments to the Author” section, enter your conflict of interest statement in the “Confidential to Editor” section, and submit your "Accept" recommendation.

Reviewer #2: All comments have been addressed

2. Is the manuscript technically sound, and do the data support the conclusions?

Reviewer #2: Yes

3. Has the statistical analysis been performed appropriately and rigorously? 

Reviewer #2: Yes

4. Have the authors made all data underlying the findings in their manuscript fully available?

Reviewer #2: Yes

5. Is the manuscript presented in an intelligible fashion and written in standard English?

Reviewer #2: Yes

6. Review Comments to the Author

Reviewer #2: Dear Authors,

I have carefully reviewed your manuscript titled "Information Technology Capability, Open Technological Innovation, and Firm Growth," and I appreciate the effort you have put into investigating this important topic.

7. PLOS authors have the option to publish the peer review history of their article (what does this mean?). If published, this will include your full peer review and any attached files.

Reviewer #2: No

---

## [Editor Report · Acceptance letter]

16 Oct 2023

PONE-D-22-14268R2 

Information technology capability, open technological innovation and firm growth 

Dear Dr. Yao:

I'm pleased to inform you that your manuscript has been deemed suitable for publication in PLOS ONE. Congratulations! Your manuscript is now with our production department. 

Kind regards, 

on behalf of

Dr. Iván Barreda-Tarrazona 

Academic Editor

PLOS ONE